biomathematics/health and disease and epidemiology/biocomplexity

epidemic resurgence, COVID-19, population dynamics, fluctuations

**Authors for correspondence:**
Jose M. G. Vilar
e-mail: j.vilar@ikerbasque.org
Leonor Saiz
e-mail: lsaiz@ucdavis.edu

# Ascertaining the initiation of epidemic resurgences: an application to the COVID-19 second surges in Europe and the Northeast United States

Jose M. G. Vilar[1,2] and Leonor Saiz[3]

[1]Biofisika Institute (CSIC, UPV/EHU), University of the Basque Country (UPV/EHU), PO Box 644, 48080 Bilbao, Spain
[2]IKERBASQUE, Basque Foundation for Science, 48011 Bilbao, Spain
[3]Department of Biomedical Engineering, University of California, 451 E. Health Sciences Drive, Davis, CA 95616, USA

JMGV, 0000-0003-4037-0746; LS, 0000-0002-6866-9400

Assessing a potential resurgence of an epidemic outbreak with certainty is as important as it is challenging. The low number of infectious individuals after a long regression, and the randomness associated with it, makes it difficult to ascertain whether the infectious population is growing or just fluctuating. We have developed an approach to compute confidence intervals for the switching time from decay to growth and to compute the corresponding multiple-location aggregated quantities over a region to increase the precision of the determination. We estimated the aggregate prevalence over time for Europe and the northeast United States to characterize the COVID-19 second surge in these regions during year 2020. We find a starting date as early as 3 July (95% confidence interval (CI): 1–6 July) for Europe and 19 August (95% CI: 16–23 August) for the northeast United States; subsequent infectious populations that, as of 31 December, have always increased or remained stagnant; and the resurgences being the collective effect of each overall region with no location, either country or state, dominating the regional dynamics by itself.

## 1. Introduction

Identifying a potential resurgence of an epidemic outbreak is crucial to timely implementation of measures for its mitigation and control. A major challenge, however, is the high uncertainty present because of the low prevalence values at which it typically happens after a long regression, as has been observed in many

locations through the ongoing COVID-19 pandemic [1–4]. At the field level, direct characterization through randomized testing would need large population studies to provide significant results and using infection case data is dependent on varying testing rates [2,3]. More robust approaches based on death counts are also affected by the extremely small number of random events on which they rely for inference [2,4]. This uncertainty in assessing the state of the outbreak for a potential resurgence is a source of delays in the decision making and intervention implementation processes.

Here, we address two main computational needs to precisely characterize a resurgence. The first one is how to establish confidence intervals in the timing of the resurgence. These intervals range from the time it is certain that the infectious population has stopped decreasing to the time it is certain that the population has started to increase with a given confidence level. The second need is how to aggregate different local data into supra-local quantities to identify whether the resurgence is a collective regional effect and, if so, to determine the initiation of the resurgence more confidently.

We focus explicitly on Europe and the northeast United States (USA), which have experienced a second surge of the COVID-19 outbreak after a similar initial outburst and subsequent regression. None of these resurgences was widely expected nor anticipated [5,6]. Both regions display high mobility among their locations and broad independence among locations to enact measures to mitigate the propagation of the outbreak. In the case of Europe, Schengen Area countries allow for unrestricted border crossings among them. Mobility restrictions, lockdowns and other non-pharmaceutical interventions were able to achieve a major regression of the outbreaks, but the gradual lifting of restrictions has resulted in a resurgence across locations in these two regions [2,7]. The characterization of the similarities and differences of the outbreak progression in these two areas is needed to provide insights into the effectiveness of the actions taken, to ascertain the extent their results can be extrapolated from one region to another, and to informedly mitigate the current and potentially forthcoming resurgences.

## 2. Methods

### 2.1. Upper and lower bounds of the growth rate determine the confidence interval of the resurgences

We consider the estimated infectious population of the specific location at time $t$ denoted by $n_I(t)$ and dynamics given by

$$\frac{\mathrm{d}}{\mathrm{d}t}n_I(t) = k_G(t)n_I(t), \tag{2.1}$$

where $k_G(t)$ is its *per capita* growth rate with upper and lower bounds of the confidence interval (CI) denoted by $k_G^U(t)$ and $k_G^L(t)$, respectively.

In epidemiology, it is customary to use the time-varying reproduction number $R_t$, which describes the expected number of infections arising from a single case in the population [8,9]. It is related to the growth rate $k_G(t)$ through the Euler–Lotka equation

$$R_t^{-1} = \int_0^\infty f_{GT}(\tau)e^{-k_G(t)\tau}\mathrm{d}\tau, \tag{2.2}$$

where $f_{GT}(\tau)$ is the probability density function of the generation time [8,9]. We consider the usual description of generation times through a gamma distribution

$$f_{GT}(\tau) = \frac{\beta^\alpha}{\Gamma(\alpha)}\tau^{\alpha-1}e^{-\beta\tau}, \tag{2.3}$$

which leads to

$$R_t = \left(1 + \frac{k_G(t)}{\beta}\right)^\alpha \tag{2.4}$$

for $k_G(t) > -\beta$ and $R_t = 0$ for $k_G(t) \leq -\beta$. The values of the parameters are given by $\alpha = \tau_G^2/\sigma_G^2$ and $\beta = \tau_G/\sigma_G^2$, where $\tau_G$ and $\sigma_G^2$ are the mean and the variance of the generation time, respectively.

The starting date of the second surge, $t_2$, is computed as the date the infectious population reached a minimum value after the maximum of the first surge at time denoted by $t_1$:

$$t_2 = \underset{t}{\arg\min}\, n_I(t) \text{ subject to } t > t_1, \tag{2.5}$$

which in continuous time corresponds to a zero value of the growth rate (reproduction number equal to 1): $k_G(t_2) \simeq 0$.

To compute confidence intervals for the switching time from negative to positive growth, we consider the confidence intervals of the population growth rate. Explicitly, when the upper bound of the growth rate is negative (reproduction number below 1), we can ascertain that the population is decreasing with a given confidence level. Analogously, when the lower bound of the growth rate is positive (reproduction number above 1), we can ascertain that the population is increasing with a given confidence level. Therefore, the lower bound, $t_2^L$, of the CI of the starting date of the second surge is computed as the last day before the minimum in which the upper bound of the CI of the growth rate is negative (reproduction number below 1):

$$t_2^L = \max_t t \text{ subject to } k_G^U(t) < 0 \text{ and } t < t_2. \tag{2.6}$$

Analogously, the upper bound, $t_2^U$, of the CI is computed as the first day after reaching the minimum in which the lower bound of the CI of the growth rate is positive (reproduction number above 1):

$$t_2^U = \min_t t \text{ subject to } k_G^L(t) > 0 \text{ and } t > t_2. \tag{2.7}$$

Note that, although the accuracy of the approach is dependent on the accuracy of the underlying characterization of the infectious population, the determination of $t_2$, $t_2^L$, $t_2^U$ is independent of potential multiplicative biases in $n_I(t)$, $k_G^U(t)$ and $k_G^L(t)$.

The approach is illustrated for Connecticut (northeast USA) and Austria (Europe's Schengen Area) in figure 1. The trajectories of the infectious populations, the growth rates, and the 95% confidence intervals (CI) for each location were downloaded on 21 April 2021, from https://github.com/Covid19Dynamics/trajectories. The data consider explicitly the age-specific infection fatality rates from Verity et al. [10], which are consistently similar among distinct locations [11,12], to infer the local infectious population from reported death counts [4]. Reproduction numbers were computed from the growth rates for each location by considering a gamma-distributed generation time, independent of the location, with a mean of 6.5 days and a standard deviation of 4.2 days [13]. The two locations in figure 1 show that, in general, there is a high uncertainty in the timing that can be attributed to the starting date of the resurgence.

## 2.2. Aggregate values provide a potential avenue to increase the reliability of the estimates for low prevalence values

The aggregate infectious population for a region is expressed as

$$n_I(t) = \sum_j n_{I,j}(t), \tag{2.8}$$

where $n_{I,j}(t)$ is the infectious population of the specific location with index $j$. Using the method of variance estimates recovery [14], which parallels the methodology of error propagation, the corresponding upper and lower confidence intervals are computed as

$$n_I^U(t) = n_I(t) + \sqrt{\sum_j (n_{I,j}^U(t) - n_{I,j}(t))^2} \tag{2.9}$$

and

$$n_I^L(t) = n_I(t) - \sqrt{\sum_j (n_{I,j}(t) - n_{I,j}^L(t))^2} \tag{2.10}$$

from the upper, $n_{I,j}^U(t)$, and lower, $n_{I,j}^L(t)$, confidence intervals for each location.

The method of variance estimates recovery cannot be used directly to compute the confidence intervals for the aggregate growth rate. We derive the expressions for the upper and lower bounds by

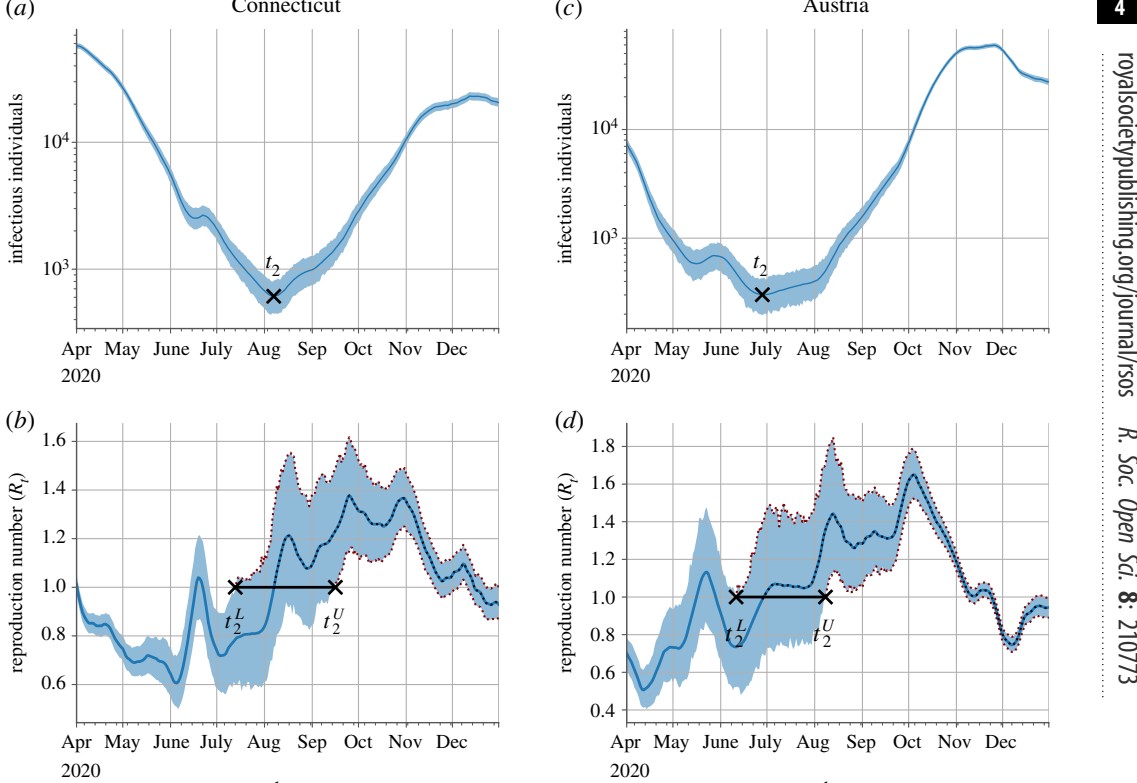

**Figure 1.** Location-specific characterization of the COVID-19 resurgence. The approach to locate the time of the resurgence and its confidence intervals is illustrated with data for Connecticut ($a,b$) and Austria ($c,d$). The top panels ($a,c$) show the temporal evolution of the infectious population with the shaded blue region indicating the 95% confidence intervals (CI). The bottom panels ($b,d$) show the temporal evolution of the reproduction number (blue line) with the shaded blue region indicating the 95% CI. The dotted lines highlight the span of the second wave and its CI over the reproduction number data. Black markers indicate the starting date of the resurgence ($t_2$) and the lower ($t_2^L$) and upper ($t_2^U$) bound of its 95% CI. Reproduction numbers were computed from growth rates considering a gamma-distributed generation interval with a mean of 6.5 days and a standard deviation of 4.2 days.

considering that the overall time-dependent growth rate is given by

$$k_G(t) = \sum_j k_{G,j}(t) \frac{n_{I,j}(t)}{n_I(t)}, \tag{2.11}$$

where $k_{G,j}(t)$ is the growth rate of the infectious population of the specific location with index $j$. This expression follows from

$$\frac{\mathrm{d}}{\mathrm{d}t} n_I(t) = \sum_j \frac{\mathrm{d}}{\mathrm{d}t} n_{I,j}(t) = \sum_j k_{G,j}(t)\, n_{I,j}(t) = k_G(t) n_I(t). \tag{2.12}$$

The corresponding upper and lower confidence intervals are computed as

$$k_G^U(t) = k_G(t) + \frac{1}{n_I(t)} \sqrt{\sum_j (k_{G,j}^U(t) - k_{G,j}(t))^2 n_{I,j}(t)^2} \tag{2.13}$$

and

$$k_G^L(t) = k_G(t) - \frac{1}{n_I(t)} \sqrt{\sum_j (k_{G,j}(t) - k_{G,j}^L(t))^2 n_{I,j}(t)^2} \tag{2.14}$$

from the upper, $k_{G,j}^U(t)$, and lower, $k_{G,j}^L(t)$, confidence intervals for each location. These expressions explicitly consider that the uncertainty in the infectious populations times the corresponding growth rate is much smaller than the uncertainty in the growth rates times the corresponding infectious population.

# 3. Results

## 3.1. The second surges started in early-mid summer with the northeast USA trailing Europe

To assess the properties of the resurgences with increased confidence, we computed the aggregate values of the infectious populations and the corresponding time-varying reproduction numbers for Europe's Schengen Area and the northeast USA from the individual values of the locations of each region [4]. We considered overall region values and overall region values excluding one location. Exclusion of one location provides an avenue to reliably infer the effects of the location in the overall region.

The initial progression of the overall infectious populations for both regions consisted of exponential growth followed by exponential decay (figure 2$a$,$b$). Subsequently, there was a sharp transition to fast exponential growth in Europe on 3 July (95% CI: 1–6 July), 2020, from an estimated infectious population of $3.0 \times 10^4$ (95% CI: $2.4 \times 10^4$–$3.5 \times 10^4$) individuals and a stagnant overall infectious population in the northeast USA, which started to grow slowly but with increasing speed on 19 August (95% CI: 16–23 August), 2020, from an estimated infectious population of $3.3 \times 10^4$ (95% CI: $2.8 \times 10^4$–$3.8 \times 10^4$) individuals.

## 3.2. The resurgence has been more abrupt and intense in Europe than in the northeast USA

Concomitantly, the time-varying reproduction numbers crossed above one on the resurgence dates less abruptly in the northeast USA than in Europe (figure 2$c$), reaching maximum values of 1.50 (95% CI: 1.48–1.51) in Europe and 1.30 (95% CI: 1.27–1.34) in the northeast USA. The sharp resurgence to exponential growth in Europe is coincidental with lifting major non-pharmaceutical interventions that curved the outbreak [15], including the coordinated end of travel bans in Schengen Area's countries on 1 July 2020 [16].

No substantial decreases in the overall infectious population, nor corresponding reproduction numbers below one, were observed for any of the two regions over three months after the starting dates of the second surges (figure 2). The estimated infectious population just stopped growing in the northeast USA in late December (figure 2$b$) and entered a prolonged stagnant state in Europe in early November (figure 2$a$).

## 3.3. Aggregate values are highly reliable compared to location-specific data

The low prevalence at the location-specific level leads to broad confidence intervals for both the infectious population and the time-varying reproduction numbers, which makes ascertaining the local growth properties of the outbreak unreliable over prolonged periods of time (figures 1 and 3, and electronic supplementary material, figure S1). The aggregated values for each region provide precise evidence of sustained growth of the outbreaks already over the summer, despite the uncertainty and variability present in each of the locations independently (figure 3 and electronic supplementary material, figure S1).

Our results also provide robust evidence that the resurgence was not driven by a unique location since any aggregate value of the starting date for each region leaving one of their locations out is within the confidence limits of the overall region (figure 3 and electronic supplementary material, figure S1). Therefore, the resurgences were the collective effect of each overall region.

# 4. Discussion

COVID-19 second surges in Europe and the northeast USA exemplify the difficulties of ascertaining the presence of an incipient epidemic resurgence and to determine whether the infectious population is growing or just fluctuating. We have provided an avenue to quantify the uncertainty present and the methodology to increase the reliability of the assessment by aggregating location-specific data in regional quantities.

The approach we have developed to quantify the uncertainty in the timing of a resurgence is based on the confidence intervals of the growth rate (or equivalently, those of the reproduction number) to ascertain, with a given confidence level, that the population is decreasing when the upper bound of the growth rate is negative and increasing when the lower bound of the growth rate is positive. The gap between these two regimes determines the CI of the minimum of the infectious population. The

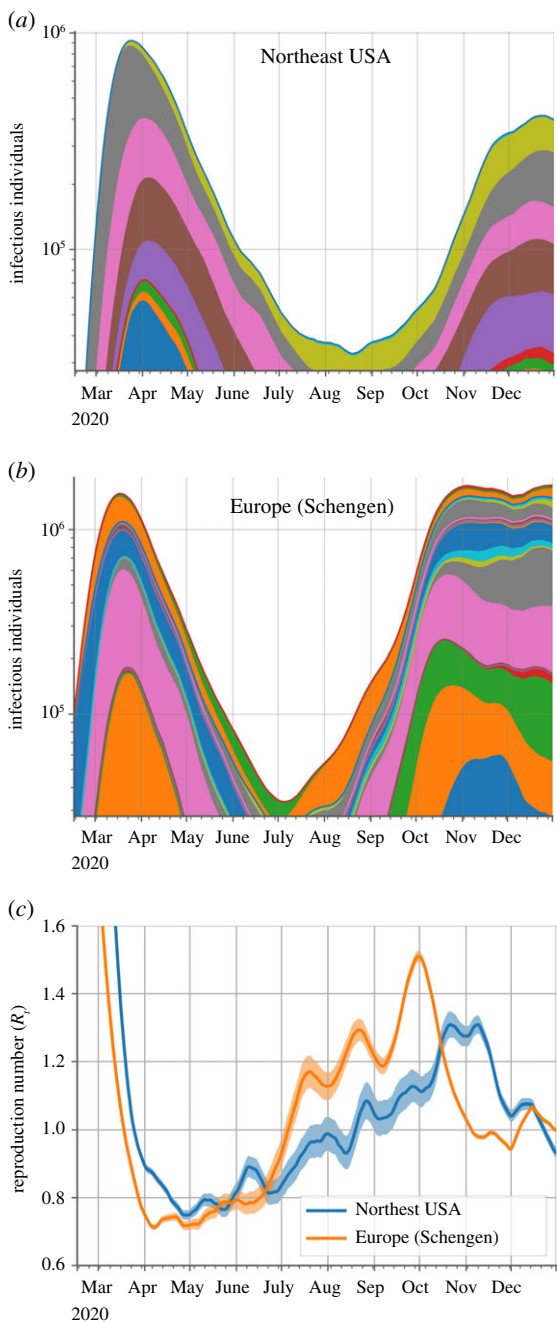

**Figure 2.** Temporal evolution of the COVID-19 outbreak in Europe and the northeast USA. Progression over time of the infectious population for countries in Europe's Schengen Area (*a*) and locations in the northeast USA (*b*) and of their corresponding reproduction numbers (*c*). Each coloured section in the area plots represents the contribution of a country (*a*) or state (*b*) to the overall infectious populations. Countries and states are arranged in alphabetical order from bottom to top. The infectious populations are plotted on a logarithmic scale to highlight the triphasic behaviour (growth-decay-growth) of the outbreak. The shaded regions in the reproduction number plots (*c*) represent the 95% CI. Locations with fewer than 30 reported COVID-19 deaths were not considered in the analysis.

aggregate values and their corresponding confidence intervals for a region, computed from those of its locations, allowed us to make precise assessments at a regional level. We obtained explicitly that regional values for the timing of COVID-19 second surges in Europe and the northeast USA are more precise than those of the individual locations and that the resurgences in these two regions are the collective effect of each overall region with no location, either country or state, dominating the regional dynamics by itself.

There are multiple behavioural, environmental and urban factors that affect the progression of infectious diseases in general and COVID-19 in particular [15,17,18]. These factors have exhibited

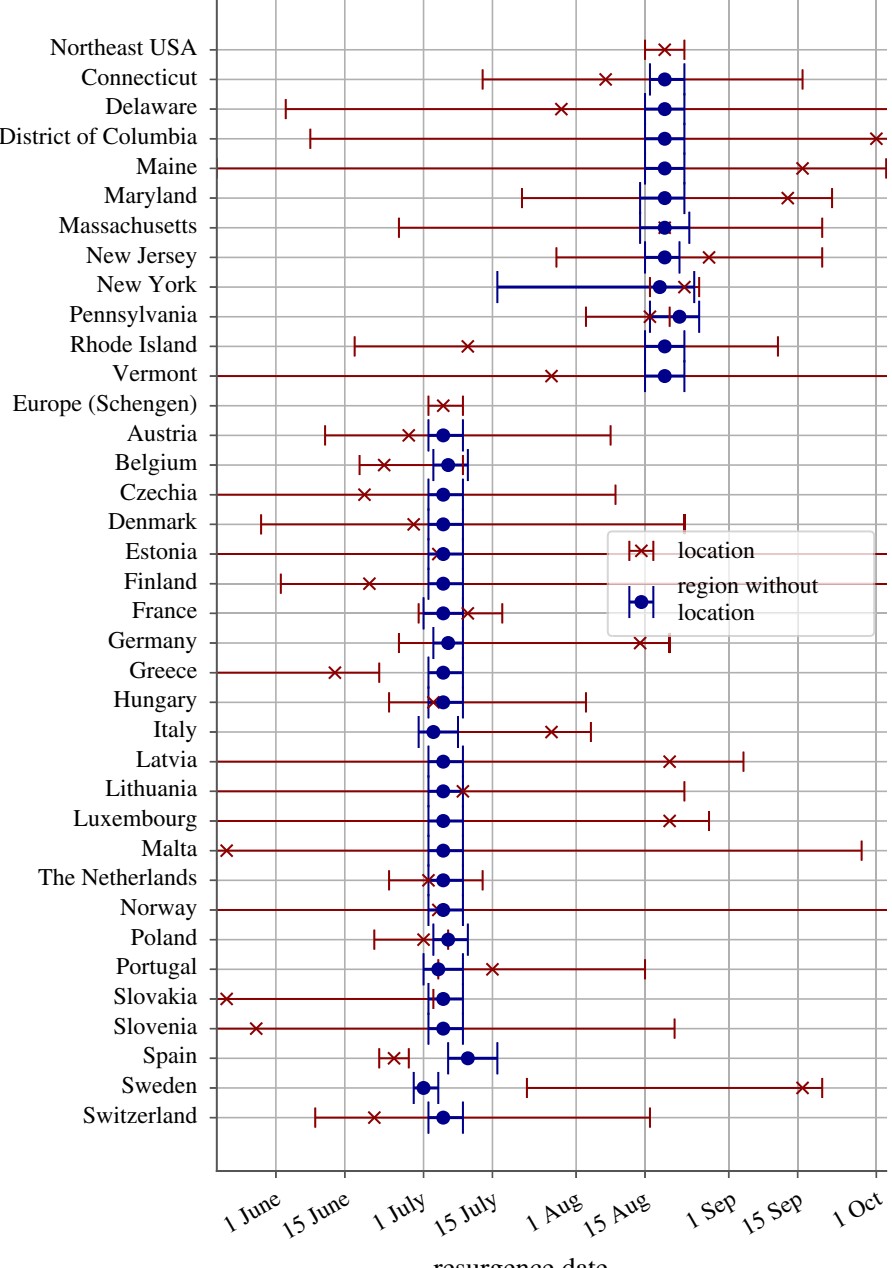

**Figure 3.** Timing of the COVID-19 outbreak resurgence in Europe and the Northeast USA. Dates of the minimum infectious population reached are shown for each location (red crosses) and for the whole region without the population of the location indicated (blue circles). The intervals represent the 95% CI. Locations with fewer than 30 reported COVID-19 deaths were not considered in the analysis.

similar patterns across states in the northeast USA and across countries in Europe's Schengen Area. Our results show that indeed the confidence intervals of the timing of the second surge largely overlap among states and among countries in these two regions.

The northeast USA, as a region, closely trailed Europe in the second surge of the outbreak, but with a markedly smaller growth and evidence of slowing down earlier in the growth phase than Europe. Key differences in the actions taken included more gradual lifting and swifter progressive reimplementation of measures in the northeast USA than in Europe [15]. Our results show, through the progression over time of the aggregate prevalence of Europe's Schengen Area countries, with high certainty, that Europe's initial acting upon the second surge in mid-late October [5] took place well after a three-month-long period of sustained growth of the COVID-19 infectious population in the overall region, which has resulted in a second surge deadlier than the first one [19]. With swifter progressive

reimplementation of measures, such a high death toll has not been reached in the northeast USA [15]. Therefore, our results highlight the need to implement policies and surveillance approaches that also include data at a supra-location level when there is high mobility among locations.

Data accessibility. This work does not include any original data. The data used in the analysis were downloaded on 21 April 2021 from https://github.com/Covid19Dynamics/trajectories, have been stored in GitHub: https://github.com/Covid19Dynamics/2ndSurges, and have been archived within the Zenodo repository: https://doi.org/10.5281/zenodo.5515775.

Authors' contributions. All authors contributed to writing the manuscript, research design, performing research, data analysis and discussion.

Competing interests. We declare we have no competing interests.

Funding. J.M.G.V. acknowledges support from Ministerio de Ciencia e Innovación under grant no. PGC2018-101282-B-I00 (MCI/AEI/FEDER, UE). L.S. acknowledges support from the University of California, Davis.

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
