## [Peer Review File · Royal Society Open Science]

Review History

RSOS-210773.R0 (Original submission)

Review form: Reviewer 1

Is the manuscript scientifically sound in its present form?

Yes

Are the interpretations and conclusions justified by the results?

Yes

Is the language acceptable?

Yes

Do you have any ethical concerns with this paper?

No

Have you any concerns about statistical analyses in this paper?

No

Recommendation?

Major revision is needed (please make suggestions in comments)

Comments to the Author(s)

See the attached file (Appendix A).

Review form: Reviewer 2

Is the manuscript scientifically sound in its present form?

No

Are the interpretations and conclusions justified by the results?

No

Is the language acceptable?

Yes

Do you have any ethical concerns with this paper?

No

Have you any concerns about statistical analyses in this paper?

No

Recommendation?

Major revision is needed (please make suggestions in comments)

Comments to the Author(s)

I like the main idea on which this manuscript is based. A reliable and robust metric is needed in the ascertainment of an [inevitably] upcoming surge/wave of cases in a region during an ongoing pandemic. That reliable metric could be a time-series of supra-location aggregated cases and/or associated R_t with its 95% confidence intervals. I think this is a good work from that point of view. Having said that, however, the manuscript in its present form is not suitable for publication at the Royal Society Open Science journal.

My comments are as follows:

The manuscript abruptly ends without any conclusion and/or main highlights.

It does not cite prior studies where needed. For example, it says it is customary to use the time-varying reproduction number without any citation.

Mathematical equations are not numbered. Although it may be a minor thing, but numbered equations, in general, prove to be helpful for potential readers in connecting them with one another.

It does not discuss the results in context or appropriately. For example, time series of aggregated infectious individuals seem to perform better than their corresponding R_t time series. I personally

do not think any practical usefulness of having Rt time-series in the ascertainment of a surge. This deserves some discussion and should have been discussed.

In addition, I find very difficult to follow the methods outlined in the paper. Which mathematical equation was used to calculate the aggregated time series of infectious individuals? What values were used for α and β ? How the growth rate was estimated? What were the mean and the variance of the generation time location-wise? And, region-wise, if applicable? There should be a parameter table describing all these things.

Decision letter (RSOS-210773.R0)

Dear Dr Vilar

The Editors assigned to your paper RSOS-210773 "Ascertaining the initiation of epidemic resurgences: an application to the COVID-19 second surges in Europe and the Northeast United States" have now received comments from reviewers and would like you to revise the paper in accordance with the reviewer comments and any comments from the Editors. Please note this decision does not guarantee eventual acceptance.

Please submit your revised manuscript and required files (see below) no later than 21 days from today's (ie 09-Aug-2021) date. Note: the ScholarOne system will 'lock' if submission of the revision is attempted 21 or more days after the deadline. If you do not think you will be able to meet this deadline please contact the editorial office immediately.

Kind regards,
Royal Society Open Science Editorial Office
Royal Society Open Science

on behalf of Professor Christine Currie (Associate Editor) and Mark Chaplain (Subject Editor)
 openscience@royalsociety.org

Associate Editor Comments to Author (Professor Christine Currie):

Comments to the Author:

Both reviewers agree that this is interesting work and are keen to see it in print. Nonetheless, they have indicated a few changes that would, we believe, improve the paper.

Reviewer comments to Author:

Reviewer: 1

Comments to the Author(s)

See the attached file.

Reviewer: 2

Comments to the Author(s)

I like the main idea on which this manuscript is based. A reliable and robust metric is needed in the ascertainment of an [inevitably] upcoming surge/wave of cases in a region during an ongoing pandemic. That reliable metric could be a time-series of supra-location aggregated cases and/or associated R_t with its 95% confidence intervals. I think this is a good work from that point of view. Having said that, however, the manuscript in its present form is not suitable for publication at the Royal Society Open Science journal.

My comments are as follows:

The manuscript abruptly ends without any conclusion and/or main highlights.

It does not cite prior studies where needed. For example, it says it is customary to use the time-varying reproduction number R_t without any citation.

Mathematical equations are not numbered. Although it may be a minor thing, but numbered equations, in general, prove to be helpful for potential readers in connecting them with one another.

It does not discuss the results in context or appropriately. For example, time series of aggregated infectious individuals seem to perform better than their corresponding R_t time series. I personally do not think any practical usefulness of having R_t time-series in the ascertainment of a surge. This deserves some discussion and should have been discussed.

In addition, I find very difficult to follow the methods outlined in the paper. Which mathematical equation was used to calculate the aggregated time series of infectious individuals? What values were used for α and β ? How the growth rate was estimated? What were the mean and the variance of the generation time location-wise? And, region-wise, if applicable? There should be a parameter table describing all these things.

===PREPARING YOUR MANUSCRIPT===

one version identifying all the changes that have been made (for instance, in coloured highlight, in bold text, or tracked changes);
 a 'clean' version of the new manuscript that incorporates the changes made, but does not highlight them. This version will be used for typesetting if your manuscript is accepted.

===PREPARING YOUR REVISION IN SCHOLARONE===

- Any electronic supplementary material (ESM).
- If you are requesting a discretionary waiver for the article processing charge, the waiver form must be included at this step.
- If you are providing image files for potential cover images, please upload these at this step, and inform the editorial office you have done so. You must hold the copyright to any image provided.
- A copy of your point-by-point response to referees and Editors. This will expedite the preparation of your proof.

- Ensure that your data access statement meets the requirements at <https://royalsociety.org/journals/authors/author-guidelines/#data>. You should ensure that you cite the dataset in your reference list. If you have deposited data etc in the Dryad repository, please include both the 'For publication' link and 'For review' link at this stage.
- If you are requesting an article processing charge waiver, you must select the relevant waiver option (if requesting a discretionary waiver, the form should have been uploaded at Step 3 'File upload' above).
- If you have uploaded ESM files, please ensure you follow the guidance at <https://royalsociety.org/journals/authors/author-guidelines/#supplementary-material> to include a suitable title and informative caption. An example of appropriate titling and captioning may be found at https://figshare.com/articles/Table_S2_from_Is_there_a_trade-off_between_peak_performance_and_performance_breadth_across_temperatures_for_aerobic_scope_in_teleost_fishes_/3843624.

Author's Response to Decision Letter for (RSOS-210773.R0)

See Appendix B.

Decision letter (RSOS-210773.R1)

Dear Dr Vilar

On behalf of the Editors, we are pleased to inform you that your Manuscript RSOS-210773.R1 "Ascertaining the initiation of epidemic resurgences: an application to the COVID-19 second surges in Europe and the Northeast United States" has been accepted for publication in Royal Society Open Science subject to minor revision in accordance with the referees' reports. Please find the referees' comments along with any feedback from the Editors below my signature.

Please submit your revised manuscript and required files (see below) no later than 7 days from today's (ie 07-Sep-2021) date. Note: the ScholarOne system will 'lock' if submission of the revision is attempted 7 or more days after the deadline. If you do not think you will be able to meet this deadline please contact the editorial office immediately.

on behalf of Professor Christine Currie (Associate Editor) and Mark Chaplain (Subject Editor)
openscience@royalsociety.org

Associate Editor Comments to Author (Professor Christine Currie):
Associate Editor

Comments to the Author:

Many thanks for addressing the referees' comments. As a suggestion for future responses to referees, it is always much easier if you rewrite the comment you are responding to in the response document before providing details of how you have addressed that comment.

I found it hard to understand the new paragraph you have added to the Discussion (P10, L32 onwards) and would suggest that this is updated before submission. In particular:

- you say that you look for a point where growth is negative before and positive after a minimum. This seems like the definition of a minimum to me and consequently sounds a little strange. You use a different wording earlier in the paper that seems to work better.
- ensure that the discussion of locations is written correctly.

===PREPARING YOUR MANUSCRIPT===

- one version identifying all the changes that have been made (for instance, in coloured highlight, in bold text, or tracked changes);
- a 'clean' version of the new manuscript that incorporates the changes made, but does not highlight them. This version will be used for typesetting.

===PREPARING YOUR REVISION IN SCHOLARONE===

- If you are providing image files for potential cover images, please upload these at this step, and inform the editorial office you have done so. You must hold the copyright to any image provided.
- A copy of your point-by-point response to referees and Editors. This will expedite the preparation of your proof.

- Ensure that your data access statement meets the requirements at <https://royalsociety.org/journals/authors/author-guidelines/#data>. You should ensure that you cite the dataset in your reference list. If you have deposited data etc in the Dryad repository, please only include the 'For publication' link at this stage. You should remove the 'For review' link.
- If you are requesting an article processing charge waiver, you must select the relevant waiver option (if requesting a discretionary waiver, the form should have been uploaded at Step 3 'File upload' above).
- If you have uploaded ESM files, please ensure you follow the guidance at <https://royalsociety.org/journals/authors/author-guidelines/#supplementary-material> to include a suitable title and informative caption. An example of appropriate titling and captioning may be found at https://figshare.com/articles/Table_S2_from_Is_there_a_trade-off_between_peak_performance_and_performance_breadth_across_temperatures_for_aerobic_scorpions_in_teleost_fishes_/3843624.

Author's Response to Decision Letter for (RSOS-210773.R1)

See Appendix C.

Decision letter (RSOS-210773.R2)

Dear Dr Vilar,

I am pleased to inform you that your manuscript entitled "Ascertaining the initiation of epidemic resurgences: an application to the COVID-19 second surges in Europe and the Northeast United States" is now accepted for publication in Royal Society Open Science.

COVID-19 rapid publication process:

We are taking steps to expedite the publication of research relevant to the pandemic. If you wish, you can opt to have your paper published as soon as it is ready, rather than waiting for it to be published the scheduled Wednesday.

This means your paper will not be included in the weekly media round-up which the Society sends to journalists ahead of publication. However, it will still appear in the COVID-19 Publishing Collection which journalists will be directed to each week (<https://royalsocietypublishing.org/topic/special-collections/novel-coronavirus-outbreak>).

If you wish to have your paper considered for immediate publication, or to discuss further, please notify openscience_proofs@royalsociety.org and press@royalsociety.org when you respond to this email.

on behalf of Professor Christine Currie (Associate Editor) and Mark Chaplain (Subject Editor)
openscience@royalsociety.org

Appendix A

Review of "*Ascertaining the initiation of epidemic resurgences: an application to the COVID-19 second surges in Europe and the Northeast United States.*"

The paper proposes a method for estimating the time of epidemics resurgence, together with confidence bounds. As the obtained confidence bounds for individual regions may be quite wide, it was found that aggregating regional infection counts can lower the uncertainty. The method was applied to the resurgence of the epidemic (second peak) in Europe and the Northeast USA. The authors obtained that the resurgence happened collectively, likely due to high mobility and coordinated implementation of measures within these areas.

I think that the presented work is interesting and relevant. Both in scientific literature and public discussions, it is commonly referred to different epidemics peaks/outbursts. However, it remains open how exactly these peaks are defined (and with what uncertainty), which is in part addressed in this paper. The paper is written clearly and succinctly. Methods and obtained results appear correct and are well explained/motivated, including a clear visual representation. Some points to be considered by the authors are the following:

- 1) Abstract: "supra-location aggregated quantities". Meaning is clear when the entire paper is read, but since the abstract should be stand-alone, it may be useful to rephrase (or better explain) these words in the abstract.
- 2) Methods, section 2.1, the first page: Reference is needed for R_t and related expressions.
- 3) Method section 2.2: Perhaps I miss something, but deriving the confidence intervals appears to be a straightforward error propagation, i.e., a variance of the sum of independent variables. The same for k_G , under the assumption that the errors for the infectious population are much lower (as also stated in the paper). Therefore, it might be useful to explain why ref. [12] (i.e., the method of variance estimate recovery) is needed?
- 4) Figure 1, in particular, the confidence intervals: The authors consider random uncertainties, which is reasonable and realistically doable. However, it might be useful to mention that the error of R_t may also depend on the systematic uncertainties in inferring the infected population. I.e., inferring the infected is model dependent, as this is not directly observable quantity (in distinction to detected case counts or deaths). These systematic uncertainties may also impact confidence bounds for R_t .
- 5) Discussion: The relevance of the work with studies to understand environmental (demographic, meteorological, etc.) factors that affect epidemics may also be useful to mention, see e.g.:
Wu et al., *Science advances*, 6(45), eabd4049.
Salom et. al., *Frontiers in Ecology and Evolution* 8, 524, 2021
An issue within the so-called ecological study design is that by considering larger (more aggregated) regions, important environmental properties may get averaged out. However, from this manuscript, it follows that analyzing regions with higher special resolution leads to larger uncertainty of the considered epidemics quantities (e.g. inferred R values). Therefore, choosing the optimal size of the regions to analyze may be (a not so easy) compromise between these two effects, where considering smaller regions (even if the data are available) may not necessarily be a better choice.
- 6) Discussion: Though the authors provide explicit examples of two geographic areas for which aggregating smaller regions is useful, the authors should more explicitly discuss by what criteria is it useful to aggregate smaller regions? For example, what if the entire USA would be aggregated (this might not be too unreasonable, as to my knowledge, there have been no travel restrictions between the federal states). It seems clear that this is not easy to address exactly/quantitatively, but some guidelines in the discussion would be useful.

Appendix B

Dear Prof. Currie,

Thank you very much for sending us the reviewer reports on our manuscript entitled “Ascertaining the initiation of epidemic resurgences: an application to the COVID-19 second surges in Europe and the Northeast United States”, Ref: RSOS-210773, by J. M. G. Vilar and L. Saiz, which we would like to resubmit for publication in *Royal Society Open Science*.

We are glad that both reviewers find our work interesting, are keen to see it in print, and suggested avenues to improve the presentation of the results. We would like to thank the reviewers for their comments and for pointing out key aspects of the manuscript that would make it more accessible to a broader audience.

A detailed response to the reviewers, indicating the changes made to the manuscript, is append below. Changes to the manuscript have been marked using track changes.

Sincerely,
Jose M. G. Vilar

Response to Reviewer 1

1) We have modified the abstract along the lines of the reviewer suggestion: “supra-location aggregated quantities” has been replaced with “multiple-location aggregated quantities over a region” and we have clarified in the last sentence of the abstract that location refers to either country or state.

2) We have added two references [Refs. 8 and 9] to provide background for R_t and related expressions.

3) We have indicated in the manuscript that the method of variance estimates recovery parallels the methodology of error propagation. The reference is included because, although the methodology is similar in both cases, errors, as expressed in terms of the variance, and confidence intervals are different quantities.

4) Following the suggestion of the reviewer, we have added the following text in page 6 of the revised manuscript: “Note that, although the accuracy of the approach is dependent on the accuracy of the underlying characterization of the infectious population, the determination of t_2 , t_2^L , t_2^U is independent of potential multiplicative biases in $n_I(t)$, $k_G^U(t)$, and $k_G^L(t)$.”

5) We have added a paragraph, including the references pointed out by the reviewer (Refs. 17 and 18), in the Discussion Section (third paragraph) to highlight the relevance of multiple behavioral, environmental, and urban factors.

6) In the newly added third paragraph of the Discussion Section, we explicitly state that behavioral, environmental, and urban factors have exhibited similar patterns across states in the Northeast US and that our results show that indeed the confidence intervals of the timing of the second surge largely overlap among states in this region. We have not considered the whole US as a region because behavioral, environmental, and urban factors vary enormously across the whole US.

Response to Reviewer 2

We have rewritten the last paragraph of the Discussion Section and added two new paragraphs (second and third paragraph of the Discussion Section) to clarify the conclusions and emphasize the highlights of our manuscript.

We have added two references [Refs. 8 and 9] to provide background for the use of the reproduction number R_t .

We have numbered the equations in the revised Manuscript.

We are not aware of the possibility of using just the time series of the infectious population to compute the confidence intervals of the time of the resurgence without using the time series of the confidence intervals of the growth rate or reproduction number. To clarify this point we have added the following text in page 5 of the revised manuscript: "To compute confidence intervals for the switching time from negative to positive growth, we consider the confidence intervals of the population growth rate. Explicitly, when the lower bound of the growth rate is negative (reproduction number below 1), we can ascertain that the population is decreasing with a given confidence level. Analogously, when the upper bound of the growth rate is positive (reproduction number above 1), we can ascertain that the population is increasing with a given confidence level."

The aggregated time series of infectious individuals was computed using equation 2.8.

The values for α and β were computed from the mean and the variance of the generation time as stated after equation 2.4.

The aggregated growth rate was estimated from the individual growth rates using equation 2.11. The individual growth rates were obtained as stated in the last paragraph of section 2.1: "The trajectories of the infectious populations, the growth rates, and the 95% confidence intervals (CI) for each location were downloaded on April 21, 2021, from <https://github.com/Covid19Dynamics/trajectories>."

We have clarified in the revised version of the manuscript that the mean and the variance of the generation time is the same for all locations: "Reproduction numbers were computed from the growth rates for each location by considering a gamma-distributed

generation time, independent of the location, with a mean of 6.5 days and a standard deviation of 4.2 days [13].”

Appendix C

Dear Prof. Currie,

Thank you very much for sending us your decision and comments on our manuscript entitled "Ascertaining the initiation of epidemic resurgences: an application to the COVID-19 second surges in Europe and the Northeast United States", Ref: RSOS-210773.R1, by J. M. G. Vilar and L. Saiz, which we would like to resubmit for publication in *Royal Society Open Science*.

We are glad that the manuscript is suitable for publication with minor changes. A response indicating the changes made to the manuscript is append below. Changes to the manuscript have been marked using tracked changes.

Sincerely,
Jose M. G. Vilar

Response to Editor

Editor's comment:

I found it hard to understand the new paragraph you have added to the Discussion (P10, L32 onwards) and would suggest that this is updated before submission. In particular:

- *you say that you look for a point where growth is negative before and positive after a minimum. This seems like the definition of a minimum to me and consequently sounds a little strange. You use a different wording earlier in the paper that seems to work better.*
- *ensure that the discussion of locations is written correctly.*

Our response:

We have rewritten the corresponding sections of the paragraph as follows: "The approach we have developed to quantify the uncertainty in the timing of a resurgence is based on the confidence intervals of the growth rate (or equivalently, those of the reproduction number) to ascertain, with a given confidence level, that the population is decreasing when the upper bound of the growth rate is negative and increasing when the lower bound of the growth rate is positive. The gap between these two regimes determines the confidence interval of the minimum of the infectious population. The aggregate values and their corresponding confidence intervals for a region, computed from those of its locations, allowed us to make precise assessments at a regional level."